# LANGUAGE AGENTS FOR HYPOTHESIS-DRIVEN CLINICAL DECISION MAKING WITH REINFORCEMENT LEARNING

**David Bani-Harouni    Chantal Pellegrini    Ege Özsoy    Nassir Navab    Matthias Keicher**

Technical University of Munich
Munich Center for Machine Learning
david.bani-harouni@tum.de

## ABSTRACT

Clinical decision-making is a dynamic, interactive, and cyclic process where doctors have to repeatedly decide on which clinical action to perform and consider newly uncovered information for diagnosis and treatment. Large Language Models (LLMs) have the potential to support clinicians in this process, however, most applications of LLMs in clinical decision support suffer from one of two limitations: Either they assume the unrealistic scenario of immediate availability of all patient information and do not model the interactive and iterative investigation process, or they restrict themselves to the limited "out-of-the-box" capabilities of large pre-trained models without performing task-specific training. In contrast to this, we propose to model clinical decision-making for diagnosis with a hypothesis-driven uncertainty-aware language agent, LA-CDM, that converges towards a diagnosis via repeatedly requesting and interpreting relevant tests. Using a hybrid training paradigm combining supervised and reinforcement learning, we train LA-CDM with three objectives targeting critical aspects of clinical decision-making: accurate hypothesis generation, hypothesis uncertainty estimation, and efficient decision-making. We evaluate our methodology on MIMIC-CDM, a real-world dataset covering four abdominal diseases containing various clinical tests and show the benefit of explicitly training clinical decision-making for increasing diagnostic performance and efficiency. Our code is available at https://github.com/dharouni/LA-CDM.

## 1 INTRODUCTION

In modern healthcare systems, accurate and efficient diagnosis stands as a critical component in patient management and treatment (Savioli et al., 2022). To form a diagnosis, clinicians often engage in clinical decision-making through a dynamic, iterative process, called differential diagnosis. This process involves forming hypotheses about the patient and testing those hypotheses through requesting and interpreting information from relevant available diagnostic tests (Sox et al., 2024). At the beginning multiple diagnoses may be possible and there is still high uncertainty. Through diagnostic testing the uncertainty is minimized and the space of possible diseases reduced until a sufficient confidence is reached, a diagnosis can be given, and treatment can begin (Sox et al., 2024). It is especially important in complex, high-stakes environments such as emergency departments, where often little is known about a patient at admission and fast and accurate diagnosis is paramount (Savioli et al., 2022).

Large Language Models (LLMs), with their ability to synthesize complex textual information, seem well-suited for aiding clinicians in this task, as most medical information is textual or can be represented in text, e.g., clinical notes, imaging reports or numerical laboratory results. This enables great flexibility and variety in medical modalities. In the medical domain LLMs have already shown great success in passing medical license exams (Singhal et al., 2023; Gilson et al., 2023) and diagnosing case challenges (Buckley et al., 2023). The potential to revolutionize healthcare through accurate diagnostic capabilities is enormous, however, existing approaches often either (1) assume immedi-

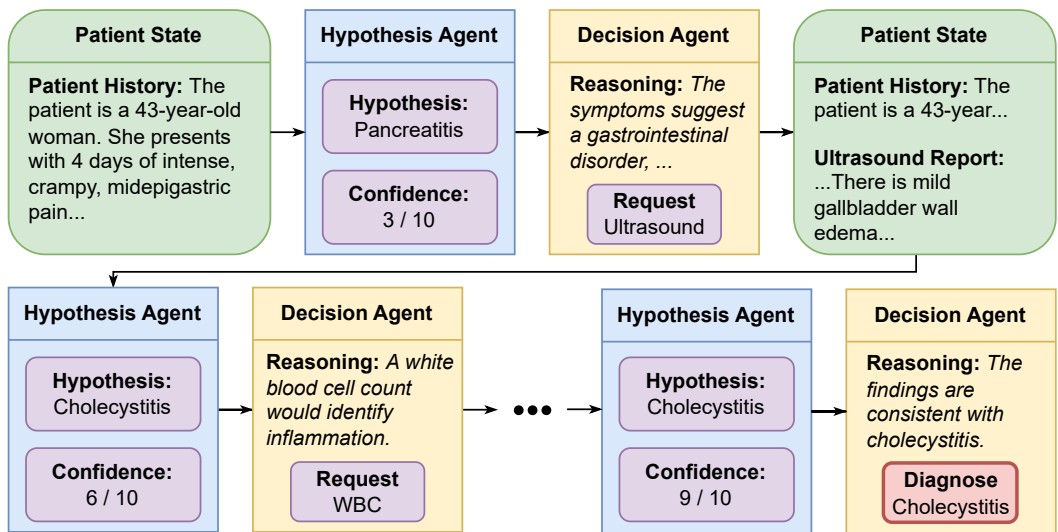

Figure 1: An illustrative process of clinical decision-making performed by LA-CDM. At the beginning, only the patient history including symptoms and family history is known. In a cyclic process, the hypothesis agent forms an uncertainty-aware hypothesis and the decision agent decides on a clinical action (request a test or diagnose). If a test is requested the results are added to the known patient information. The cycle repeats until a final diagnosis is given.

ate availability of all patient data (Buckley et al., 2023; McDuff et al., 2023; Chen et al., 2025), which is rarely the case in practice, or (2) rely on the often limited "out-of-the-box" behavior of pre-trained LLMs without any task-specific fine-tuning to the complexities of diagnostic decision-making (Hager et al., 2024; Li et al., 2024; Nori et al., 2025). This mismatch between research and real-world clinical decision-making limits the applicability of LLMs to the clinical setting.

In this paper, we address the above limitations by modeling and training Language Agents for Clinical Decision Making (LA-CDM), tasked with iteratively reducing hypothesis uncertainty through repeated diagnostic testing. Inspired by cognitive research on human clinical decision-making (Sox et al., 2024), we design a two-agent system replicating the two main cognitive tasks of clinicians involved in clinical decision-making. It consists of one LLM agent, the *hypothesis agent*, forming the most likely diagnosis hypothesis based on all available patient information and estimating its confidence in that hypothesis, and another agent, the *decision agent*, that evaluates the patient information and the hypothesis agent's output to either provide a diagnosis or request an additional diagnostic test (Figure 1). To train this system, we propose a novel training strategy with three distinct objectives that target the core principles of successful clinical decision-making (Sox et al., 2024):

1. **Accurate hypothesis generation:** Using supervised fine-tuning, the hypothesis agent is trained to form a correct hypothesis. Since information on the patient is only uncovered step-by-step, the agent has to make use of limited information from various data sources.

2. **Hypothesis uncertainty estimation:** Using reinforcement learning the hypothesis agent is trained to be well-calibrated in its verbalized uncertainty estimation. A well-calibrated model that is e.g. 60% certain on a hypothesis will be correct in 60% of cases.

3. **Efficient decision-making:** Using reinforcement learning, the decision agent is trained to select the most informative next test and reach a diagnosis when sufficiently confident, taking test costs into account. The model gets rewarded for a final correct diagnosis, reinforcing the testing pathway that led to that diagnosis.

Analogously to doctors graduating from medical school, LLMs have a strong medical knowledge foundation, but are not trained on performing clinical decision-making. Clinicians learn this skill through years of experience, pointing towards experience-based reinforcement learning as a prime paradigm for teaching clinical decision-making to LLMs. Further, since optimal testing pathways

are not known, supervised learning of clinical decision-making is infeasible. The interplay of the three objectives results in the model learning which tests to request in order to increase the hypothesis confidence leading to an accurate diagnosis at minimal testing cost. This guides the model to request those tests that are most informative in a given situation. To the best of our knowledge, we propose the first method for explicitly training LLMs for clinical decision-making.

We evaluate our method on MIMIC-CDM (Hager et al., 2024), a real-world dataset focused on four abdominal diseases that mirrors clinical workflows by modeling differential diagnosis through sequential test requests. Its standardized test naming and inclusion of lab results, notes, and imaging reports make it uniquely suited for training and evaluating LLM-based diagnostic reasoning. Our experiments demonstrate the benefit of explicitly training clinical decision-making. Notably, training reduces the diagnostic cost, which has practical implications in reducing healthcare costs, diagnosis time, and patient discomfort (Savioli et al., 2022). We show the benefit of our hypothesis-driven approach to clinical decision-making and demonstrate that the model adapts its testing procedure to the patient at hand, placing this work as a step towards patient-specific personalized differential diagnosis.

## 2 RELATED WORK

**Reinforcement Learning for Clinical Decision Making**  Reinforcement learning has been explored for cost-efficient clinical decision-making based on tabular data. Yu et al. (2023) train SM-DDPO, a model that iteratively requests laboratory tests optimizing diagnostic performance and cost-efficiency. Their method features an imputation model, estimating missing (and not yet requested) laboratory tests and a classification model predicting the diagnosis. A policy network trained with Q-learning predicts the next action, i.e. which test to request or which diagnosis to give. They show improvements in efficiency at a similar diagnostic performance compared to baselines making use of all available information. However, as the method is only compatible with tabular data, it neglects many important medical modalities, like clinical notes or imaging reports, which are often crucial for diagnosis.

ED-Copilot (Sun et al., 2024) employs a Language Model for encoding serialized patient laboratory values which is trained end-to-end with two MLPs, one predicting test requests, and one predicting the task of outcome severity. They train their method in two stages. First, they use supervised learning to teach the model to predict all tests in a pre-defined order. In a second training stage, they use reinforcement learning to finetune the model to reduce time cost of the testing regiment. They also report great reduction in testing time, while remaining similar in performance to baselines making use of all available information. While this method uses a language model as its encoding backbone, this language model is not directly requesting tests. Same as the previous method, they further only consider tabular laboratory values and therefore do not make use of valuable information in textual form.

Reflexion (Shinn et al., 2023) addresses general LLM decision-making by introducing a zero-shot reinforcement learning approach. Rather than fine-tuning the LLM with traditional reinforcement learning, the method allows the model to iteratively attempt the same task. After each trial, a limited number of previous attempts and a numerical reward are added to the input context, guiding the next generation. While this is effective at general decision-making, this approach cannot be applied in the clinical context, where multiple diagnosis trials with the same patient and intermediate diagnosis correctness feedback are infeasible.

**LLMs for Clinical Decision Making**  LLMs have so far only been used as zero-shot methods for clinical decision support. Hager et al. (2024) place large "out-of-the-box" language models in an evaluation framework where they are tasked with interactively requesting diagnostic tests and diagnose patients. They show severe limitations of LLMs for clinical decision-making and report worse diagnostic performance than clinicians. Vaid et al. (2024) approach clinical decision-making with a tool-using LLM. Through zero-shot prompting, they provide the LLM with a number of available tools, e.g., a symptom tool for retrieving the symptoms, or a imaging study tool for getting any imaging reports on the patient. They evaluate various proprietary LLMs, with GPT-4 (Achiam et al., 2023) showing the best performance. Liu et al. (2024) model LLM clinical decision-making as a multi-agent setting, where a doctor agent communicates with a patient agent who can detail his symptoms and a technician agent who can perform laboratory or imaging results. They compare

different prompting techniques, like chain-of-thought (Wei et al., 2022) or one-shot prompting and show that GPT-4o achieves the best performance. All these methods do not attempt to improve model performance by explicitly training clinical decision-making with LLMs.

## 3 PRELIMINARIES

### 3.1 REINFORCEMENT LEARNING ON LARGE LANGUAGE MODELS

Reinforcement learning has emerged as a powerful framework for fine-tuning LLMs by aligning their behavior with desired objectives, especially when no clear ground truth for this behaviour exists. Most notably, Reinforcement Learning with Human Feedback (RLHF) (Ouyang et al., 2022) was used to align LLM generations with human preferences. LA-CDM utilizes a direct reinforcement learning method based on Group Relative Policy Optimization (GRPO) (Shao et al., 2024) without requiring human feedback.

Let $\mathcal{X}$ be the space of textual inputs (prompts) and $\Theta \subseteq \mathbb{R}^d$ be the parameter space of a LLM. We denote by $\pi_\theta(a \mid s)$ the stochastic policy of the LLM, parameterized by $\theta$, which specifies the probability distribution over a discrete vocabulary $\mathcal{A}$ (the set of possible tokens) given a state $s \in \mathcal{S}$. In the context of language modeling, a state $s$ is typically a sequence of tokens $(x_1, \ldots, x_t)$ consisting of a prompt and previously generated tokens. An action $a$ is the next token to be generated, yielding the updated state $s\prime = (x_1, \ldots, x_t, a)$. Every state transition gives rise to an reward the model is trained to maximize.

### 3.2 CONFIDENCE CALIBRATION OF LARGE LANGUAGE MODELS

While LLMs have shown impressive capabilities in many language-related tasks, hallucinations and confidently-presented wrong answers are a common and well-known problem (Kadavath et al., 2022). A well-calibrated model is able to express confidence that aligns with the epistemic probability of correctness. This means that of all the answers which are presented with a confidence of $0 \leq p \leq 1$, the fraction of correct answers is $p$.

In this work, we train confidence calibration with reinforcement learning as proposed by Stangel et al. (2025). They model confidence calibration as a betting game, where the model bets on the correctness of its answer. If it is correct with a high confidence it receives a large reward. However, if it is wrong with a high confidence, the punishment becomes large. Analogously, if the answer is wrong the model receives the largest reward if it expresses a low confidence. Concretely, they use the reward function

$$R(y_{pred}, c, j) = \begin{cases} \log(c), & \text{if } J(y_{pred}) \text{ is True} \\ \log(1 - c), & \text{if } J(y_{pred}) \text{ is False}, \end{cases}$$

where $y_{pred}$ is the predicted answer, $0 < c < 1$ is the (scaled and clipped) confidence prediction, and $J(\cdot)$ is a binary function evaluating the correctness of $y_{pred}$. The reward is then scaled to be between $-1$ and $1$. They train the model using Proximal Policy Optimization (PPO) (Schulman et al., 2017). This training approach removes the need for an artificially constructed ground truth confidence dataset, as done by other confidence calibration methods (Azaria & Mitchell, 2023; Kadavath et al., 2022), and instead only requires a measure of answer correctness. The authors prove that an optimal policy under their reward design produces perfectly calibrated confidence expressions.

## 4 LANGUAGE AGENTS FOR CLINICAL DECISION MAKING

We propose LA-CDM consisting of two language agents, hypothesis agent and decision agent, trained with three different objectives. The hypothesis agent is trained in accurate hypothesis generation through supervised fine-tuning and uncertainty-awareness through reinforcement learning. The decision agent is trained in decision-making using reinforcement learning. Both agents share the LLM weights, so training one agent also influences the other. In Figure 2, we show the full model. The two agents and the three training objectives will be explained in detail in this section.

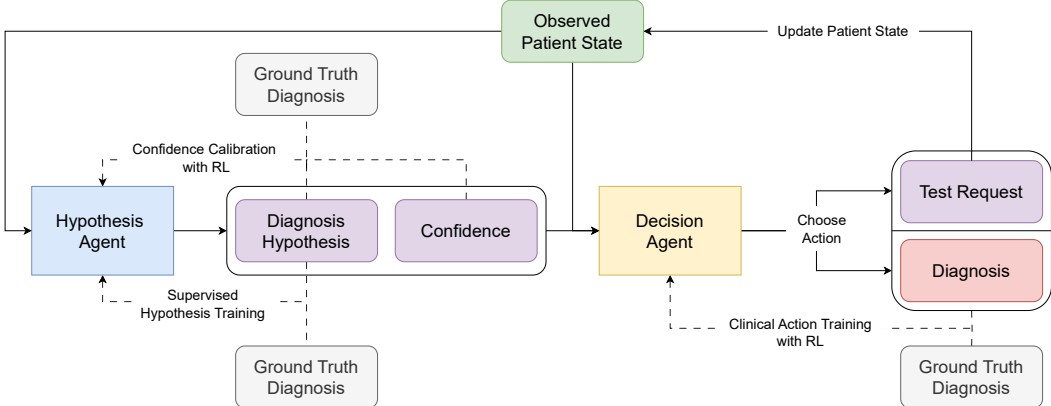

Figure 2: Overview of our method LA-CDM and its three training objectives. The hypothesis agent receives the current patient state and predicts a hypothesis and confidence. The hypothesis generation is trained supervised, the confidence calibration using reinforcement learning. The hypothesis agent output and the current patient state are then provided to the decision agent that is trained to decide on an optimal clinical action (test request or diagnosis) using reinforcement learning.

## 4.1 MODELING CLINICAL DECISION MAKING

**Clinical Decision Making Environment** In order to train our model in decision-making it has to operate in a reinforcement learning environment to explore testing strategies and receive reward signals. The model learns decision-making through interacting with this environment and diagnosing patients. Let each patient be described by a number of $n$ test results $[t_i]_{i=1}^n$ as textual records of clinical notes, imaging reports and laboratory panels. Since patient information is iteratively uncovered through the model's test requests $t_j$, we define the currently observed patient state at time-step $j$ as the set of all observed tests and write it as $p_j$. The correct diagnosis for the patient is denoted by $y_{true}$.

As we simulate a clinical patient-doctor interaction, $p_0$, the initial observed patient state, consists of the first clinical notes detailing symptoms, medical and family history. This information is always available to the model. The environment advances step-wise with each model action. If the model requests an additional test, the observed patient state is updated, and the results are made available to both the hypothesis agent and the decision-making agent for the next step. The simulation ends when the model provides a diagnosis for the patient or if one of the two failure cases is reached: (1) the model exceeds the specified maximum number of generated tokens, or (2) the model violates its specified output format.

**Hypothesis Agent** Through its system prompt, the hypothesis agent is introduced to its task and provided with the possible diagnoses. At each time-step $j$ of the environment, the hypothesis agent $\mathcal{H}$ is given the currently observed patient state $p_j$ to predict the most likely diagnosis $h_j$ based on the limited available information, as well as the confidence in that prediction $c_j$. It therefore produces a mapping

$$\mathcal{H} : p_j \rightarrow \{h_j, c_j\}.$$

The model generates this output in a format of "Hypothesis: $h_j$, Confidence: $c_j$". The agent reports numerical confidence estimations on a scale of 0 to 10, where 10 means absolute certainty of the correctness of the hypothesis, and 0 means absolute certainty that it is incorrect.

**Decision Agent** The decision agent $\mathcal{D}$ is the actor advancing the environment. It decides on which action to take at each time-step. Through its system prompt it is provided with its task, a list of tests present in the dataset, and the possible patient diagnoses. Provided with the currently observed patient state $p_j$ and the hypothesis agent's hypothesis $h_j$ and confidence $c_j$, it produces a decision on whether to request another diagnostic test $t_j$ and move on to the next time-step $j + 1$ or whether to commit on a specific diagnosis $y_{pred}$ for the patient and end the episode. Formally, it produces a

mapping

$$\mathcal{D} : \{p_j, h_j, c_j\} \rightarrow \begin{cases} t_j & \textit{if a further test is requested} \\ y_{pred} & \textit{if a diagnosis is given} \end{cases}$$

Specifically, we employ the ReAct prompting technique (Yao et al., 2022), to prime the model to first produce a reasoning trace, following chain-of-thought principles (Wei et al., 2022), and then provide action and action input (in our case, the specific test or diagnosis) in a structured format.

If a further test was requested, the test results are appended to the conversation context as a user response to the LLMs generation. Since we can only work with retrospective data, where not every test result is present for every patient, we cannot always fulfill the model's request. In these cases, the user reply tells the model the requested test is unavailable and asks it to choose a different action. The implications of this will be discussed later on. Also if the model requests tests or provides diagnoses that are not on the list of possible tests or, respectively, diseases, the model is asked to choose a different action.

## 4.2 TRAINING CLINICAL DECISION MAKING

In our training objectives we follow the three main principles of clinical decision-making, as proposed by Sox et al. (2024): (1) accurate hypothesis generation, (2) hypothesis uncertainty estimation, and (3) efficient decision-making. The first two objectives are trained with respect to the generations of the hypothesis agent, the last objective is trained on the environment interactions of the decision agent. We follow a cyclic training approach, where each objective is trained individually for a specified number of episodes after which the objective changes to the next one until the cycle repeats, resulting in much more stable training compared to optimizing all objectives simultaneously.

**Training Hypothesis Generation** A baseline of good clinical decision-making is a high accuracy in hypothesis generation. If the model knows the most likely candidate for the diagnosis, it can adapt its testing strategy to quickly confirm or reject this hypothesis. While the model interacts with the environment, the hypothesis agent is confronted with various patient states consisting of different subsets of test combinations. We collect all contexts shown to the model within all episodes of a patient batch, usually including multiple hypothesis generation steps per patient. To perform supervised fine-tuning, we create target sequences, consisting of the collected conversation contexts concatenated with the correct hypothesis generation $y_{true}$. We compute the cross-entropy loss for the sequences, ignoring the token at the position where the model should place its confidence score.

**Training Uncertainty-Awareness** Since the available patient information is often limited, especially at early stages of the diagnostic process, and the available data does not always clearly point at a specific diagnosis, uncertainty is inherent to clinical decision-making. An accurate intrinsic estimation of that uncertainty by the model can give it an improved basis for decisions on when to stop the diagnostic process and produce a diagnosis. In this work, we train confidence calibration following a method proposed by Stangel et al. (2025) as previously introduced in Section 3.2, however instead of PPO, we use the GRPO algorithm (Shao et al., 2024). We define our correctness measure $J(h_j)$ as equality between the predicted hypothesis $h_j$ and the ground truth diagnosis $y_{true}$.

**Training Efficient Clinical Action Selection** At the core of training clinical decision-making lies the training of clinical action selection. During interaction with the clinical decision-making environment the model can freely choose to iteratively request any number of tests in any order. Given the vast number of possible diagnostic pathways, defining an optimal test sequence for each patient is infeasible, we therefore do not have a ground truth on which tests to perform. To still be able to train clinical decision-making, we propose to use reinforcement learning, where the model can learn through trial-and-error which tests are useful in which situations. Equal to the confidence calibration training, we employ the GRPO algorithm (Shao et al., 2024) for reinforcement learning training. Through interaction with the clinical decision-making environment the model freely requests different tests until it decides on a diagnosis. The requested tests are provided to the model by the environment whenever available. If not available the model is notified and asked to request a different test. We design our diagnosis reward function $R_{diag}$ to present the model with a fixed positive reward $r_{pos}$ if the final diagnosis at the end of the diagnosis episode was correct, or a fixed negative reward $r_{neg}$ if it is wrong. Additionally, we punish the model with reward $r_{invalid}$ if the

model violates the specified format. Our reward function is thus:

$$R_{diag}(y_{pred}) = \begin{cases} r_{pos} & \textit{if } y_{pred} = y_{true} \\ r_{neg} & \textit{if } y_{pred} \neq y_{true} \\ r_{invalid} & \textit{for out-of-format generations} \end{cases}$$

Not just diagnosis accuracy but also diagnostic efficiency is an important factor in clinical decision making. Different tests have different costs for the healthcare system, e.g., a CT scan is much more expensive, than a simple blood value. The model should therefore only request expensive tests, when necessary for diagnosis. To incentivize efficient behaviour, we add an additional reward function $R_{cost}$ that punishes test usage relative to their cost:

$$R_{cost}(T) = -\sum_{t_j \in T} c(t_j),$$

where $T$ is the list of all performed tests and $c(t_j)$ is the cost of test $t_j$.

The interaction of these three objectives enables the model to learn which tests to request to enhance hypothesis confidence, ultimately leading to a more accurate diagnosis while also taking test cost into account. This drives the model to prioritize tests that provide the most informative insights in a given situation.

## 5 EXPERIMENTAL SET-UP

**Dataset and Pre-Processing** We evaluate our method on the MIMIC-CDM dataset (Hager et al., 2024), a curated subset of MIMIC-IV (Johnson et al., 2020) designed for modeling sequential clinical decision making. It contains 2,400 patients diagnosed with one of four abdominal conditions: appendicitis, cholecystitis, diverticulitis, or pancreatitis. This focused setting on four pathologies reflects real clinical workflows, where physicians perform differential diagnosis with a narrowed down space of possible diseases and request tests to distinguish between likely candidates. The dataset includes patient histories (symptoms, comorbidities, family histories) and physical exam notes for all patients. It also provides 5,959 textual imaging reports (CT, x-ray, ultrasound, MRI) and 143,191 lab results (blood, urine, microbiology), however, not every test result is reported for every patient. If multiple values for a specific test were recorded during the hospital stay of the patient only the first one was included in MIMIC-CDM to simulate an early diagnosis after hospital admission. Crucially, the dataset includes comprehensive mappings of test names across patients, an essential feature for modeling test requests reliably. Without this normalization, a model could not query the same test across different cases due to inconsistent naming in clinical documentation. To our knowledge, MIMIC-CDM is the only publicly available dataset that enables simulation of this setting. For our use-case, we construct the set of available tests as: physical examination, all imaging modalities, and the most common laboratory panels. These panels are collections of individual tests that are usually ordered together.

**Metrics** To evaluate model performance, we report class-wise accuracies along with their mean, as well as micro and macro F1-scores. Additionally, we compute the Expected Calibration Error (ECE) to assess confidence calibration of our hypothesis agent. ECE measures the discrepancy between predicted confidence scores and actual accuracy. A lower ECE indicates better model calibration, meaning the predicted probabilities align well with actual correctness frequencies.

**Baselines** We compare our method to various zero-shot and trained baselines. In their work, Hager et al. (2024) introduce both the MIMIC-CDM dataset and evaluate how various "out-of-the-box" pre-trained models perform when tasked with clinical decision-making. We compare with OASST, the best performing model from this evaluation, however, a direct comparison is very difficult for multiple reasons. First, as a zero-shot method, the model was evaluated on the complete dataset, whereas we reserve some part of that dataset for training. Second, the clinical decision framework was constructed differently, as the OASST model is not provided with the possible diagnosis classes or available tests in its prompt. Additionally, as an approximate upper-bound for our method, we compare with a Qwen-2.5-7B-Instruct model (Yang et al., 2024) identical to our LLM backbone,

Table 1: Performance comparison of LA-CDM and baseline methods. We report class-wise accuracies and F1-scores. The avg. test cost is the mean cost of the tests for each patients in the test set. *OASST results are taken from Hager et al. (2024). It is evaluated on a different test set in a different framework. $^{\dagger}$SM-DDPO can only process tabular data. ZS = zero-shot

| Method | Accuracy | | | | | F1-Score | | Avg. |
| | Append. | Cholec. | Divert. | Pancr. | Mean | Micro | Macro | Test Cost |
| --- | --- | --- | --- | --- | --- | --- | --- | --- |
| OASST* | 82.0 | 48.0 | 45.5 | 44.1 | 54.9 | - | - | - |
| SFT-all | 98.4 | 89.8 | 95.8 | 87.5 | 92.8 | 93.6 | 92.9 | $3792.79 |
| SM-DDPO$^{\dagger}$ | 74.3 | 0.0 | 15.6 | 58.0 | 37.0 | 45.4 | 31.8 | - |
| ReAct | 90.2 | 79.7 | 66.7 | 62.9 | 74.9 | 79.1 | 74.8 | $1480.32 |
| LA-CDM (ZS) | 73.5 | 55.1 | 72.0 | 57.5 | 64.5 | 65.3 | 64.5 | $1521.73 |
| LA-CDM | **93.1** | **83.6** | **75.0** | **73.5** | **81.3** | **84.1** | **81.3** | **$1295.61** |

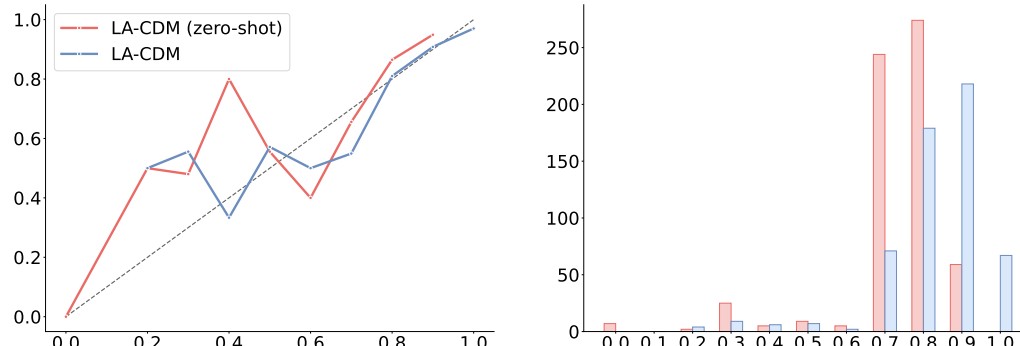

Figure 3: *Left:* Calibration curves before and after training LA-CDM. *Right:* Distribution of confidence estimations before and after training LA-CDM.

trained with supervised fine-tuning to predict the correct diagnosis. Instead of requesting tests it simply receives all the available patient information directly which is not a realistic diagnostic process. We refer to this method as SFT-all. We compare to three other adaptive test selection methods on MIMIC-CDM. SM-DDPO (Yu et al., 2023) trains a MLP using reinforcement learning for clinical decision making, however, the method is only able to process tabular data and can only request laboratory values. ReAct (Yao et al., 2022) is a zero-shot decision making method and LA-CDM (ZS) is the untrained version of our method.

## 6 RESULTS AND DISCUSSION

**Comparison with Baselines** Our comparison with baselines is shown in Table 1. When comparing with OASST (Hager et al., 2024), LA-CDM shows improvement in accuracy for each class resulting in almost 30 percentage points difference when comparing the mean of all class accuracies. The performance shows the largest improvement for pathologies that are less common compared with the majority class appendicitis. While the methods are not directly comparable as outlined in section 5, the high performance improvement clearly shows the advantage of training on the task of clinical decision-making and not relying on inherent capabilities of large pre-trained models. This is further supported by the evaluation results of ReAct and of LA-CDM as a zero-shot model. In comparison to our trained model, we see a great performance improvement achieved through training. Importantly, the untrained methods also diagnoses less efficiently than the trained version, acquiring, in the case of ReAct, almost $200 more test cost to form a diagnosis. Also the individual performance of the hypothesis agent benefits from training LA-CDM. Through our hypothesis generation training, we improve the ability of the model to form correct hypotheses from 75.7% to 81.9%. We equally show an improvement of uncertainty-awareness through training the confidence calibration objective. The ECE decreases from 0.069 to 0.037. We visualise the calibration curves and

Table 2: Ablation of the inclusion of the test cost reward function.

| Test Cost | Accuracy | | | | | F1-Score | | Avg. Test Cost |
|---|---|---|---|---|---|---|---|---|
| | Append. | Cholec. | Divert. | Pancr. | Mean | Micro | Macro | |
| | **93.5** | 81.4 | **82.1** | 72.3 | **82.3** | **84.3** | **82.4** | $1427.85 |
| ✓ | 93.1 | **83.6** | 75.0 | **73.5** | 81.3 | 84.1 | 81.3 | **$1295.61** |

Table 3: Ablation of the hypothesis-driven approach. Experiments without test cost. HA = hypothesis agent, DA = decision agent.

| Agents | | Accuracy | | | | | F1-Score | | Avg. Test Cost |
|---|---|---|---|---|---|---|---|---|---|
| HA | DA | Append. | Cholec. | Divert. | Pancr. | Mean | Micro | Macro | |
| | ✓ | 93.0 | 79.7 | 79.2 | 62.2 | 78.5 | 81.7 | 78.6 | **$1410.01** |
| ✓ | ✓ | **93.5** | **81.4** | **82.1** | **72.3** | **82.3** | **84.3** | **82.4** | $1427.85 |

confidence distribution of the two models in Figure 3, where the trained model shows better calibration, especially at often predicted confidences. When comparing with SM-DDPO, the other trained baseline, we can see the benefit of being able to process textual input. SM-DDPO almost entirely fails as it is limited to tabular data which is not sufficient in many diagnostic tasks like the one at hand, where imaging results are paramount. The SFT-all model serves as a rough upper bound, as it leverages all available retrospective patient data, an unrealistic setup for real-time clinical decision-making. Therefore, it cannot be used for direct patient interactions. Naturally, the performance is very strong, however, as all available tests are used, the diagnostic cost is more than tripled. The substantial reduction in test cost highlights the efficiency of our approach. Moreover, we observe evidence of patient-adaptive testing strategies aligning with best practices: for suspected cholecystitis, the model most frequently selects ultrasound (64.9% of cases), the gold-standard test (Hirota et al., 2007); for appendicitis, it prioritizes CT scans (85.1% of cases), consistent with diagnostic guidelines (Di Saverio et al., 2020). These results demonstrate that our method not only achieves high diagnostic accuracy but also optimizes resource usage in a clinically meaningful way. The reasoning traces generated by chain-of-thought prompting further enable the interpretation of the model's testing pathways. We report qualitative examples of the model's generations in Appendix B.

**Ablation Study** In Table 2, we observe that while the inclusion of the test cost reward function, performs similarly to training without test cost, the average test cost per patient is reduced significantly. This shows how the model learns to become more efficient in its testing pathways while achieving similar diagnostic accuracy. More information on the choice of test costs is given in appendix C. Further, we evaluate the benefit of our hypothesis-driven approach in Table 3. When removing the hypothesis agent from our methodology, the decision agent has to learn to request tests and to diagnose without relying on the uncertainty-aware hypothesis generation capabilities of the hypothesis agent, explicitly trained for these objectives. The benefit of the hypothesis agent is demonstrated clearly by improvements in all metrics.

**Limitations** The data we are training on is retrospective with different tests missing for different patients. Furthermore, the available tests are those tests that the clinicians involved in treating that patient performed. The model can only explore a limited spread of testing pathways. It can therefore only learn to become more efficient within the testing protocols performed by doctors, e.g by skipping unnecessary test. Our objective is therefore not to discover novel clinical strategies, but to optimize decision-making within realistic, expert-demonstrated behavior. Simulation of unavailable test data could open up a pathway for modeling a more holistic clinical decision-making environment.

## 7   Conclusion

In this work, we propose a novel approach to clinical decision-making using LLMs by explicitly modeling the iterative hypothesis refinement process that clinicians follow in practice. Unlike previous work that either assumes full access to patient data or relies on untrained, out-of-the-box LLM behavior, our method introduces a structured, two-agent system trained to dynamically acquire and integrate diagnostic information. By leveraging reinforcement learning to optimize uncertainty estimation and decision-making, we enable a model that not only improves diagnostic accuracy but also enhances efficiency in medical testing. Our evaluation on the MIMIC-CDM dataset demonstrates that our approach surpasses existing baselines, achieving more accurate diagnoses with fewer diagnostic tests. This reduction in testing needs has direct benefits for real-world healthcare settings, including lower costs, faster diagnosis times, and reduced patient burden. Furthermore, our findings highlight the ability of the model to adapt its testing strategy based on patient-specific information, an essential step toward personalized AI-assisted healthcare.

## Acknowledgements

The authors gratefully acknowledge the financial support by the Bavarian Ministry of Economic Affairs, Regional Development and Energy (StMWi) under project ThoraXAI (DIK-2302-0002), and the German Research Foundation (DFG, grant 469106425 - NA 620/51-1).

## Reproducibility Statement

In order to ensure reproducibility, we describe implementation details in Section 5 and Appendix C. Further, Appendix A provides the exact prompts used in our experiments. Lastly, we we will publish our complete code containing preprocessing scripts, model code, training pipelines, and evaluation scripts, as stated in the abstract.

## Ethics Statement

The development of LA-CDM represents a step forward in AI-driven clinical decision support, offering a structured and hypothesis-driven approach to diagnosis. By iteratively requesting and interpreting clinical tests, LA-CDM has the potential to improve diagnostic accuracy while optimizing resource utilization, ultimately enhancing patient outcomes. From an ethical perspective, the deployment of LA-CDM necessitates rigorous safeguards to ensure patient safety, fairness, and transparency. As an AI system designed to support medical decision-making, it is crucial that its recommendations remain interpretable and aligned with established medical guidelines. Moreover, biases present in training data must be carefully monitored to prevent disparities in diagnostic performance across different patient populations. In the broader societal context, LA-CDM has the potential to support clinicians in environments with high cognitive load, such as emergency departments and primary care settings, where time-sensitive decision-making is critical. However, it is essential that such AI-driven tools are positioned as augmentative rather than substitutive, ensuring that they enhance rather than diminish the role of healthcare professionals. Additionally, accessibility must be prioritized to ensure equitable deployment across diverse healthcare systems, including under-resourced settings.

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

# APPENDIX

## A PROMPTS

### A.1 HYPOTHESIS AGENT PROMPT

```
You are a helpful medical expert.  Your task is to create a hypothesis for
a diagnosis of a patient admitted to the emergency department complaining
about abdominal symptoms.  You can hypothesise one of the following four
conditions:

appendicitis
cholecystitis
diverticulitis
pancreatitis

You have to base your hypothesis on the information on the patient that
will be provided to you at the end.  If you are uncertain, provide your
best guess.
Additionally, you always have to provide your confidence as a number from
0, 1, 2, 3, 4, 5, 6, 7, 8, 9, or 10, where 0 means very uncertain and 10
means very certain.

####
Always use the following format for your hypothesis:

Hypothesis:  Here, you put your chosen condition as a single word.  Nothing
else goes here.
Confidence:  Here, you put your confidence as a number between 0 and 10

####
Here are the four master rules:
1.  Only answer in the above format!
2.  Always provide a hypothesis, even if you are uncertain and always
provide a confidence.
3.  Only give one of the four possible conditions and use their exact
naming.
4.  Do not answer with anything other than the hypothesis and confidence as
specified in the format description.

Any violation of these rules, will have dire consequences for the patient.
Any correct hypothesis will be greatly rewarded.

####
Here is the available information on the patient:
{provided_test_results}
```

### A.2 DECISION AGENT PROMPT

```
You are an interactive, helpful medical expert.  Your task is to diagnose
patients admitted to the emergency department complaining about abdominal
symptoms into the following four conditions:

appendicitis
cholecystitis
```

```
diverticulitis
pancreatitis
```

To aid you in your diagnosis, you can request the results of the following
tests for the patient:

```
Physical Examination
CT
MRI
Radiograph
Ultrasound
Complete Blood Count
Basic Metabolic Panel
Comprehensive Metabolic Panel
Renal Function Panel
Liver Function Panel
Urinalysis
Electrolyte Panel
```

At the start you will be provided with some context on the patient.
At the beginning of every step you will be further provided with a current
hypothesis of the condition.

```
####
```
In your diagnosis process always use the following format:

```
Thought:  [Here, you should think about what to do next.]
Action:  [Here, you can either write "Diagnosis" if you are confident
enough to provide a diagnosis or "Test" if you want to request a single
test.]
Action Input:  [If you chose "Diagnosis" as your action, you should put
the correct condition of the four provided above here.  You MUST put one
of the four conditions here.  If you chose "Test", provide the test name
from the list mentioned above.  You can only request one single test per
action.  Use the exact naming of the test as given above.  Do not add any
specification, abbreviation or explanation to the test name.]
Observation:  [Here, you will receive the test results for the patient.
Always write the keyword "Observation:" when requesting a test.]
---
The current hypothesis is:  [Here you will see the current hypothesis]
(confidence [Here, you will get the confidence in the hypothesis on a scale
of 0 (very uncertain) to 10 (very certain)])

Thought:  [Here, you evaluate the test result given above and continue with
your next thought and action]
Action:  [You can request more tests with the action "Test".]
Action Input:  [...]
Observation:  [...]

---
The current hypothesis is:  ...  (confidence:  ...)
...
```

```
---
The current hypothesis is ...  (confidence:  ...)

Thought:  [Finally, when you are confident in a diagnosis you provide your
reasoning for that diagnosis first.]
Action:  [You can provide a diagnosis with the action "Diagnosis"]
Action Input:  [Here, you provide the condition you choose from the four
given above.]

####
If you requested a test, the user will provide you with the test results
if available.  If the test is not available, do not request the same test
again.  If you receive an unknown test or diagnosis error, try again and
ensure that you use the exact wording given above.

####
Here are the six master rules:
1.  Ask for exactly one test per test action.
2.  Only ask for one of the tests given in the list above.
3.  If a test returns as "not available" under no circumstance request the
same test again.  Try some other test.  Ask for no test twice.
4.  Use the exact naming of the tests as given in the list above
without additional specifications of the region, test abbreviations, or
explanations.
5.  Be brief and concise in all your text.
6.  Keep asking for tests until you are sufficiently confident but always
provide a valid diagnosis in the end.  Make sure to ask for enough tests to
reach a diagnosis.

Any violation of these rules, will have dire consequences for the patient.
Any correct diagnosis will be greatly rewarded.

####
Here is some context on the patient:
{patient_history}
```

## B    QUALITATIVE EXAMPLES

We show exemplary LA-CDM decision-making procedures. In Figure 4 and Figure 5, we show two successful predictions. In Figure 6, we show a failure case where the model gives a diagnosis prematurely without providing a reason, probably due to an overly confident hypothesis.

## C    IMPLEMENTATION DETAILS

**Dataset Pre-processing**    The initial patient history is shortened by processing them with a Mixtral-8x7B (Jiang et al., 2024) model, prompted to summarize the most important aspects. Importantly, this summarization is performed without any auxiliary patient information, such as disease labels. Also the imaging reports are shortened, wherever a separation into sections was available, by only keeping the findings section of the report and removing the remaining sections. The original dataset does not have a data split, since it was intended for the evaluation of zero-shot models. We therefore split the data into a training set of 80%, and a validation and test set of 10% each.

**Training Details**    We use a PyTorch implementation of the Qwen-2.5-7B-Instruct model Yang et al. (2024) as base LLM and fine-tune it using LoRA Hu et al. (2022). We train our method in a

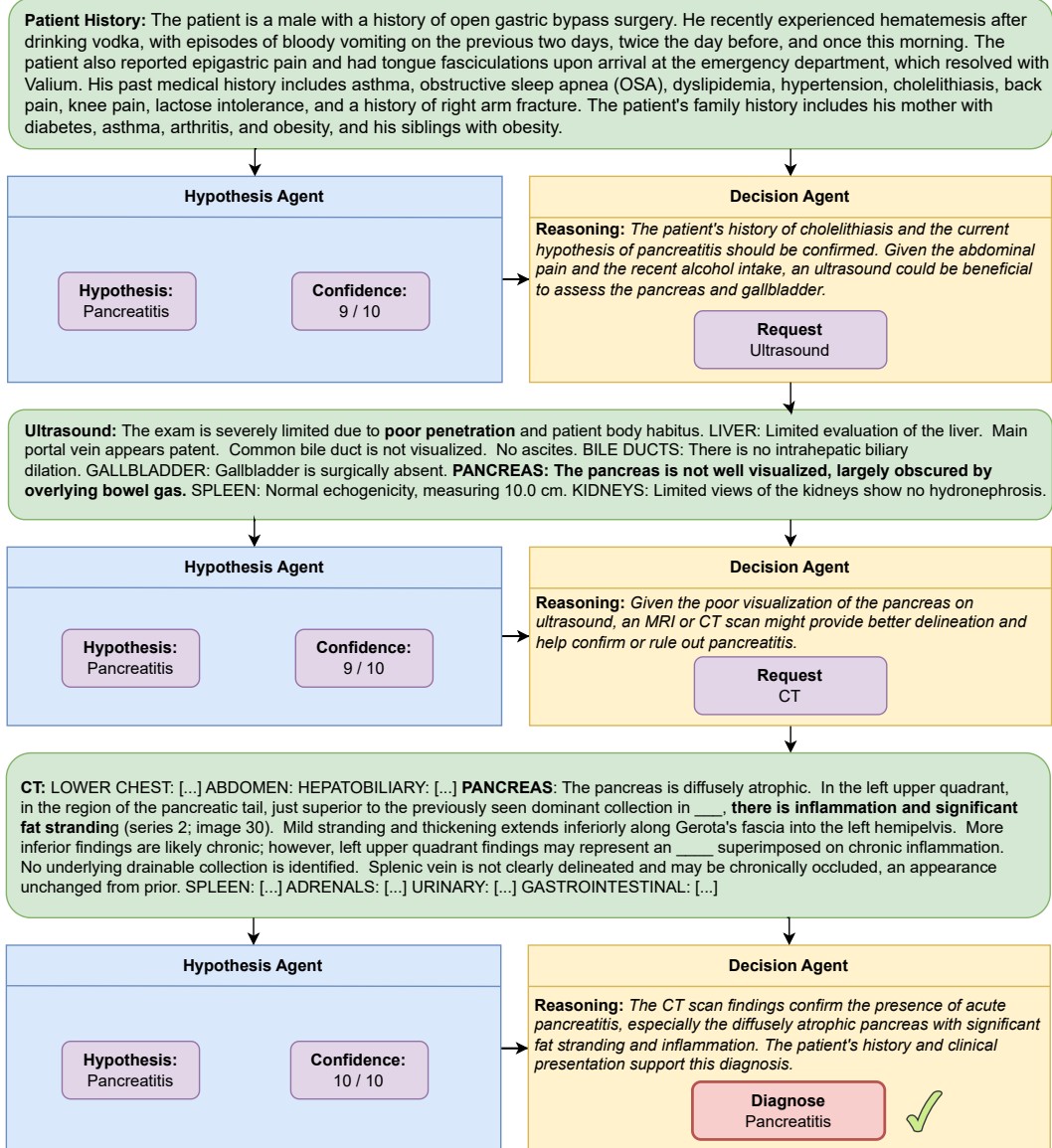

Figure 4: A successful diagnosis performed with the evaluation of ultrasound and CT. The model decides to confirm an initial suspicion with an ultrasound, however, the results are not conclusive, prompting another imaging test.

cyclic training regiment with each objective trained for 100 steps in the order of (1) clinical action training, (2) hypothesis generation training, and (3) confidence calibration training. All objectives are optimized using the Adam optimizer with a learning rate of $1e-5$ and a batch size of 2 on a single NVIDIA A40 GPU until convergence on the validation set, taking approximately 3 days.

**Test Costs** Specific cost data was taken from the 2025 standard charges table sourced under the CMS HHS price transparency rule from the Beth Israel Deaconess Medical Center, where the MIMIC-CDM dataset was collected. The specific test costs are given in Table 4. For calculating the specific test punishment within the test cost reward function, all costs where normalized to sum to the reward for a correct diagnosis, such that a diagnosis pathway that requires all tests achieves a total reward of zero. Thus, it is always better to request more tests if it helps the model to reach the correct diagnosis.

**Patient History:** The patient is a person identified as _ M, experiencing a 4-day persistent **right-sided abdominal pain** that has intensified, with no associated GI symptoms or diarrhea, unlike past experiences. Notably, they recently completed a course of Z-pak for an ear infection. Their medical history includes a **diagnosis of Crohn's disease** at a certain age, though they are not currently on any medication for it, having previously been on Asacol for an extended period. Their physical and social histories are unremarkable, as is their familial medical history. The recent abdominal pain, occurring outside of a context of GI symptoms or diarrhea, is a concern given the patient's history of Crohn's disease.

**Hypothesis Agent**

**Hypothesis:** Appendicitis

**Confidence:** 8 / 10

**Decision Agent**

**Reasoning:** *Given the persistent right-sided abdominal pain and the initial hypothesis of appendicitis, a CT scan would be the most informative test to confirm or rule out this diagnosis, given the patient's history of Crohn's disease.*

**Request** CT

**CT:** The visualized lung bases are clear with no focal consolidation or pleural effusion. The visualized heart and pericardium are unremarkable. The liver, gallbladder, spleen, pancreas, and bilateral adrenal glands are within normal limits. Both kidneys enhance and excrete contrast symmetrically without evidence of hydronephrosis or renal calculi. There is no free air or free fluid within the abdomen. **The appendix appears fluid filled with mild mucosal hyperemia and transverse diameter of approximately 12 mm with surrounding periappendiceal stranding.** There is mild thickening at the base of the cecum. There is no free air within the abdomen. Trace free fluid is noted within the pelvis. Retroperitoneal and mesenteric lymph nodes do not meet CT size criteria for pathologic enlargement. The bladder, distal ureters, rectum, and sigmoid colon are unremarkable. The visualized osseous structures show no focal lytic or sclerotic lesion suspicious for malignancy.

**Hypothesis Agent**

**Hypothesis:** Appendicitis

**Confidence:** 10 / 10

**Decision Agent**

**Reasoning:** *The CT scan confirms the presence of a fluid-filled appendix with surrounding periappendiceal stranding and mild mucosal hyperemia, which strongly supports the diagnosis of appendicitis. The patient's history of Crohn's disease does not rule out appendicitis, and the imaging findings are consistent with this condition.*

**Diagnose** Appendicitis ✓

Figure 5: A short example diagnosis. In many cases, the model learns to correctly predict the condition very briefly with just a CT.

Table 4: Test costs as derived from the standard charges table at the Beth Israel Deaconess Medical Center.

| Test Name | Cost in USD |
| --- | --- |
| CT | 1306 |
| MRI | 4866 |
| Radiograph | 434 |
| Ultrasound | 1288 |
| Complete Blood Count | 71 |
| Basic Metabolic Panel | 298 |
| Comprehensive Metabolic Panel | 636 |
| Renal Function Panel | 394 |
| Liver Function Panel | 413 |
| Urinalysis | 50 |
| Electrolyte Panel | 134 |

## D    EXPERIMENTS ON COST SENSITIVITY

In order to evaluate cost sensitivity, we perform two experiments: (i) Inter-institutional cost variation, and (ii) sensitivity to large price deviations.

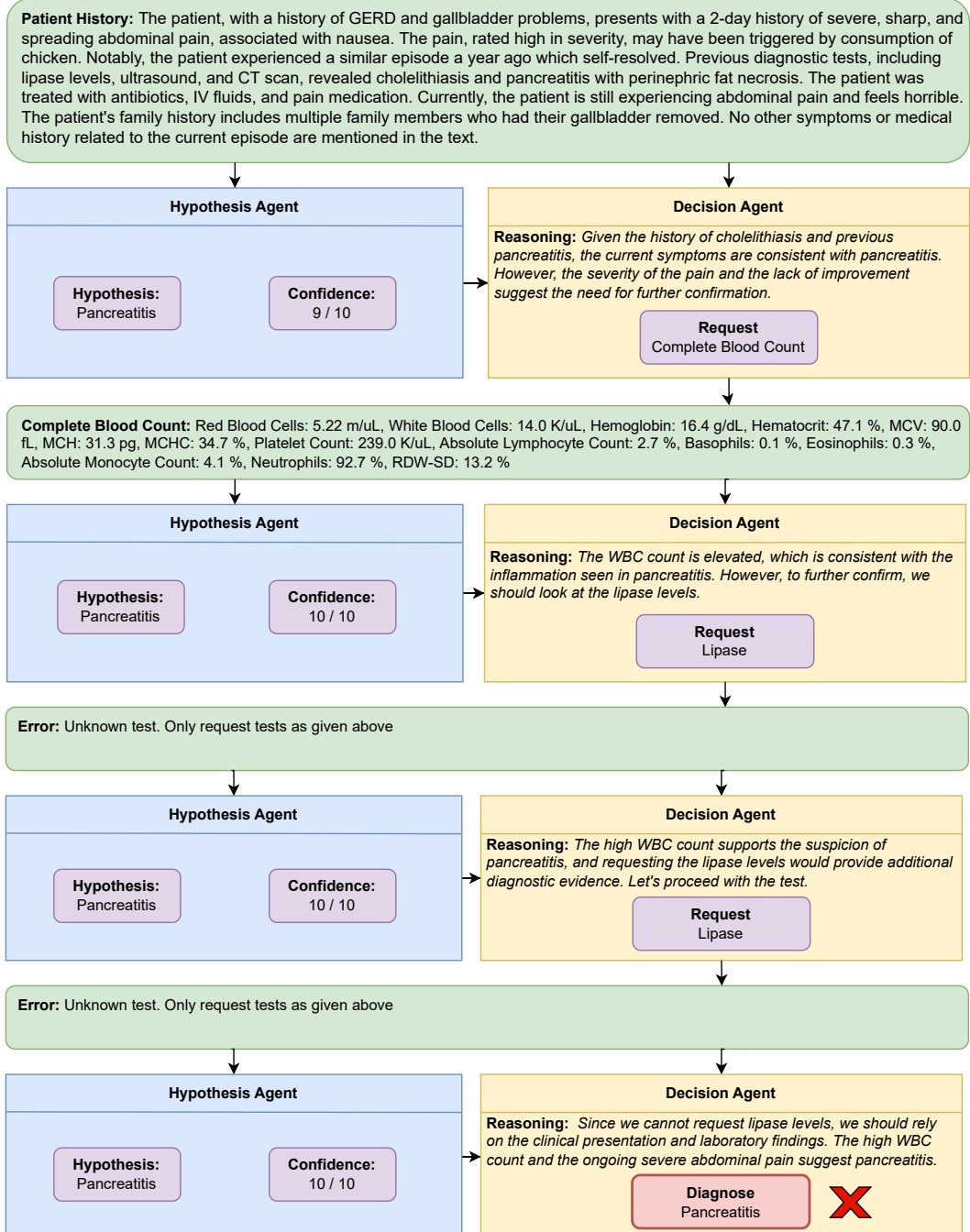

Figure 6: An example of a failed diagnosis of cholecystitis. The model reaches a high confidence in a wrong diagnosis too quickly and requests non-existent tests.

## D.1 INTER-INSTITUTIONAL VARIATION

To evaluate sensitivity to inter-institutional cost variation, we create a new cost table that exhibits slight variations in pricing to our original cost table. This simulates a cost table from a different institution with different prices that, however, stay within a similar range. We provide the changed cost table in Table 5. After retraining, we observe that both diagnostic performance and test cost efficiency remain largely unaffected. The results are shown in Table 6.

Table 5: Test costs variations are derived as slight deviations from the standard charges table at the Beth Israel Deaconess Medical Center.

| Test Name | Cost in USD | Cost after variation in USD |
|---|---|---|
| CT | 1306 | 1325 |
| MRI | 4866 | 4891 |
| Radiograph | 434 | 407 |
| Ultrasound | 1288 | 1274 |
| Complete Blood Count | 71 | 66 |
| Basic Metabolic Panel | 298 | 290 |
| Comprehensive Metabolic Panel | 636 | 626 |
| Renal Function Panel | 394 | 397 |
| Liver Function Panel | 413 | 418 |
| Urinalysis | 50 | 53 |
| Electrolyte Panel | 134 | 136 |

Table 6: Performance comparison of LA-CDM with and without test cost variation. (CV) marks the model with trained with the test cost variation.

| Method | Accuracy | | | | | F1-Score | | Avg. |
|---|---|---|---|---|---|---|---|---|
| | Append. | Cholec. | Divert. | Pancr. | Mean | Micro | Macro | Test Cost |
| LA-CDM | **93.1** | **83.6** | 75.0 | 73.5 | 81.3 | **84.1** | 81.3 | $1295.61 |
| LA-CDM (CV) | 93.0 | 75.0 | **83.6** | **77.8** | **82.3** | 83.9 | **82.3** | **$1276.22** |

### D.2 SENSITIVITY TO SPECIFIC PRICE CHANGES

In order to test model reactance to specific cost signals, we construct an artificial test table where one test's cost, the complete blood count (CBC), has been increased to be very expensive. In this setting all requests of the CBC, lead to a test cost punishment added to the final reward of $-10$. This is significantly larger than all other test costs which are normalized to be in the range of $0$ to $1$. After training with this cost table, we observe the number of CBC requests going down significantly compared to evaluating the model trained with the standard cost table. The prevalence is reduced from $19.2\%$ of cases to $2.5\%$ of cases. The diagnostic performance is also reduced as CBC can be an informative test in many situations. The F1 micro score decreases from $84.1$ to $81.9$ and the F1 macro score decreases from $81.3$ to $77.8$. This experiment clearly demonstrates the sensitivity to specific test cost distributions.

## E USE OF LARGE LANGUAGE MODELS

We employed ChatGPT to enhance the clarity of the manuscript by focusing on grammar corrections, shortening overly complex sentences, and providing alternative wording suggestions. All outputs were manually reviewed before inclusion, and no new technical material, code, results, or figures were generated by the tool.

