# OpenReview forum: "Language Agents for Hypothesis-driven Clinical Decision Making with Reinforcement Learning"
_ICLR.cc/2026/Conference — ICLR 2026 Poster_

### Official Review · Reviewer_epNq · 2025-10-31

**Soundness:** 3
**Presentation:** 2
**Contribution:** 3
**Rating:** 4
**Confidence:** 4

**Summary:**

This paper introduces LA-CDM, a two-agent LLM framework for clinical diagnosis that mirrors clinicians’ workflow: a hypothesis agent proposes a diagnosis with an explicit confidence, and a decision agent either orders the next test or finalizes the diagnosis. The authors train three objectives in a cyclic schedule—(1) supervised hypothesis generation, (2) RL-based confidence calibration (a betting-style reward), and (3) RL for action selection with a test-cost penalty—so the system learns to raise hypothesis confidence while minimizing unnecessary testing. Evaluated on MIMIC-CDM (2,400 patients; four abdominal diseases; labs, notes, imaging reports), LA-CDM improves mean class accuracy to 81.3% with macro-F1 81.3 and reduces average test cost to USD1,295.61 versus a zero-shot ReAct baseline at USD1,480.32; an ablation shows the cost reward cuts cost from USD1,427.85 to USD1,295.61 with similar accuracy. Calibration also improves (ECE 0.069 → 0.037).

**Strengths:**

1. The paper's theme is based on real-world clinical problems. The design of the two agents reflects the actual diagnostic process: hypothesis formation → targeted testing → decision-making.
2.The paper demonstrates a strong focus on cost. The R_cost term explicitly penalizes expensive tests (e.g., MRI USD4,866 vs CBC USD71), driving parsimonious testing.
3.The paper has principles for handling uncertainty. Confidence calibration via RL (betting-style reward) trains the model to express calibrated confidences without ground-truth confidence labels.
4.The paper aligns with the guidelines for patient adaptive behaviors.The trained model preferentially orders ultrasound for cholecystitis (64.9%) and CT for appendicitis (85.1%), matching clinical practice.

**Weaknesses:**

1.This study's evaluation is limited to four abdominal diseases based on a single retrospective dataset; it lacks broader disease coverage or external validation. It lacks generalizability.
2.The data in the paper has limitations in retrospective constraints and test availability bias. Many requested tests are unavailable in logs, forcing the agent to try alternatives; this caps exploration to clinician-observed pathways and can bias learned policies.
3.The paper preprocesses patient information.Summarization of histories and imaging reports (e.g., keeping only findings) could remove cues clinicians use, but the impact isn’t quantified.
4.The paper did not analyze the sensitivity of the cost model.Test costs are tied to a single hospital’s 2025 charge schedule; robustness to alternative cost tables or scaling is not reported.

**Questions:**

1.How does LA-CDM generalize to other institutions/specialties or non-abdominal conditions? Any plans for external-site validation or leave-one-hospital evaluation?
2.Does summarizing histories and trimming imaging reports change diagnostic difficulty or shift feature reliance? Please provide with/without pre-processing results.
3.Do you quantify how often “test unavailable” occurs and its effect on learning/inference? Could synthetic imputation or simulation reduce bias from missing tests?
4.How sensitive are policies to the cost table (e.g., ±x% scaling, switching to a different hospital’s charges)? Please report accuracy/cost under several cost schedules.

---

> ### Author Response · Authors · 2025-11-21
> **Rebuttal by Authors**
>
> We thank the reviewer for their feedback. We appreciate their observation that the paper’s theme is grounded in “real-world clinical problems”, and that the “design of the two agents reflects the actual diagnostic process.” We also acknowledge their comments on our “strong focus on cost.” Finally, we appreciate their recognition that our model’s behavior “aligns with clinical practice.” In the following, we will address all your open questions:
>
> **1. On the dataset**
>
> The MIMIC-CDM dataset contains a subgroup of patients reporting to the emergency department with abdominal pain. It thus represents a challenging and realistic scenario, where doctors must perform differential diagnosis between a narrowed down set of diseases with overlapping symptoms. This step-wise reduction of possible diagnoses is common practice in medicine, making the dataset a meaningful proof of concept for demonstrating clinical decision-making and differential diagnosis. The challenging aspect of differential diagnosis is not differentiating between a broken bone and appendicitis but differentiating between related conditions with similar symptom presentation. Previous studies [1] of LLMs evaluated in zero-shot out-of-distribution settings have shown unsatisfactory results. We propose for the first time a formulation enabling task-specific fine-tuning of LLMs for clinical decision-making.
>
> There are no other comparable public datasets available from other hospitals to evaluate cross-hospital generalization or broader disease coverage.
>
> [1] Hager et al. "Evaluation and mitigation of the limitations of large language models in clinical decision-making." Nature medicine 30.9 (2024): 2613-2622.
>
> **2. Learning from retrospective data and possibility of data imputation**
>
> Our method is constrained to the set of clinically valid tests and observed diagnostic pathways contained in the dataset. As noted in the limitations section, the model can therefore only explore a limited spread of testing pathways and is restricted to learning efficiency improvements within testing protocols that were actually performed by doctors, e.g., by replacing or skipping unnecessary tests. In this sense, our objective is not to discover novel clinical strategies, but to optimize decision-making within realistic, expert-demonstrated behavior. We have expanded our limitations section to reflect this.
>
> Synthetic imputation of patient data could indeed be used to enable free agent exploration, however, realistic imputation of uncommon testing pathways is out-of-scope and remains future work.
>
> **3. Preprocessing of clinical notes**
>
> The summarization of clinical notes is necessary to fulfill our academic computational requirements, as overly long texts increase GPU memory use and processing speed. However, we want to stress that the summarization happens without any auxiliary patient information, most importantly without providing the patient disease labels. We have clarified this in our revised manuscript in appendix C. Any lost information, which is bound to occur in any summarization, therefore only leads to underestimating the performance of our method compared to if full clinical notes could be used. Due to computational constraints, we cannot evaluate our method in a setting where full clinical notes are used.
>
> **4. Sensitivity to cost variations**
>
> We thank the reviewer for suggesting this interesting evaluation. In order to evaluate cost table sensitivity, we performed two experiments.
>
> 1) *Inter-institutional cost variation*
>
> To evaluate sensitivity to institutional cost variations, we simulate a potential other hospital’s cost distribution by performing slight variations to our cost table. After these variations, the model’s performance and cost efficiency remains largely unaffected. Details of this experiment can be seen in the appendix of the revised manuscript.
>
> 2) *Sensitivity to specific test cost differences*
>
> In order to test model reactance to a different cost signal, we construct an artificial test table where one test’s cost, the complete blood count, has been increased to be very expensive. After retraining with this adapted cost table, we observe that the test is requested far less often. Under normal prices, the test is requested in 19.1% of cases, when it is more expensive this rate reduces to 2.5% of cases.
>
> Details on both experiments are provided in the appendix D of the revised manuscript.
>
> **5. Prevalence of unavailable tests**
>
> In approximately 43% of generations, a requested test is not available. The model is designed to work with unavailable tests and simply requests valid alternatives in these cases, demonstrating robustness against missing data.
>
> Thank you again for your feedback. We hope our reply clarifies your open questions and will be helpful for your final evaluation.

---

> > ### Comment · Reviewer_epNq · 2025-11-28
> > **Thank you for the detailed response and additional experiments**
> >
> > I thank the authors for their detailed response and for conducting additional experiments during the rebuttal period.
> >
> > Regarding the cost sensitivity: I particularly appreciate the effort in adding the sensitivity analysis (Appendix D). The experiment showing that the model reduces requests for an artificially expensive test (from 19.1% to 2.5%) effectively demonstrates that the policy is learning to optimize cost rather than just mimicking patterns. The inter-institutional cost variation experiment also alleviates my concern about robustness.
> >
> > Regarding the dataset and limitations: I acknowledge the authors' point regarding the scarcity of public medical datasets for cross-hospital validation. While the limited scope (four abdominal diseases) remains a constraint for broad generalization, I understand this is a common challenge in the field. The clarification on why imputation was out of scope and the details on test unavailability (43%) are helpful.
> >
> > Regarding preprocessing: The explanation regarding computational constraints and the label-agnostic nature of the summarization is reasonable.
> >
> > However, my primary concern regarding the evaluation scope remains. While I understand the difficulty in obtaining public cross-hospital datasets, the evaluation being limited to a single retrospective dataset with only four abdominal diseases still limits the evidence for the method's generalizability and broader clinical utility.
> >
> > I have adjusted my review to reflect the resolved questions (cost sensitivity and preprocessing), but I will maintain my original score as I believe the limited empirical scope is a significant hurdle for acceptance.

---

> ### Author Response · Authors · 2025-11-28
>
> Thank you for your detailed response. We appreciate your constructive feedback and are happy that we were able to address most of your concerns. Your suggested changes and additions helped us further strengthen the paper.
>
> We agree that the evaluation scope is limited, however, this dataset is the only publicly available dataset for the important and realistic use case of iterative diagnosis. We agree with you that the scarcity of medical datasets is a problem inherent to most works in this field. We believe that research in this field is still valuable as it offers proof-of-concepts that also guide the community towards the collection of more comprehensive datasets.

---

### Official Review · Reviewer_j7xL · 2025-11-01

**Soundness:** 3
**Presentation:** 3
**Contribution:** 3
**Rating:** 6
**Confidence:** 3

**Summary:**

This paper proposes **LA-CDM (Language Agents for Clinical Decision Making)** — a two-agent LLM system for **iterative, hypothesis-driven diagnosis**.
A **Hypothesis Agent** generates provisional diagnoses with calibrated confidence, while a **Decision Agent** chooses whether to request additional tests or finalize a diagnosis.
Training uses a **hybrid pipeline**:
1. **SFT** on labeled cases for hypothesis quality.
2. **Reinforcement Learning (GRPO)** for calibrating confidence via a “betting” reward.
3. **RL** for cost-aware decision optimization, balancing diagnostic accuracy and test cost.

The environment is based on **MIMIC-CDM**, including text notes, imaging summaries, and lab data for 2,400 cases across four abdominal conditions.
Results show LA-CDM achieves **higher accuracy and lower average testing cost** than ReAct or untrained baselines, with **better calibration (ECE ↓ from 0.069 to 0.037)**.

**Strengths:**

1. **Innovative design:** Two-agent division closely reflects clinical reasoning loops.
2. **Calibration reward:** Improves reliability of verbal confidence estimates.
3. **Cost-sensitive optimization:** Demonstrates efficiency improvements without sacrificing accuracy.
4. **Transparent methodology:** Prompts, cost tables, and training configs are public.
5. **Multimodal context:** Integrates notes, labs, and imaging text within a unified environment.
6. **Clear ablations:** Each module’s contribution is empirically validated.

**Weaknesses:**

1. **Limited dataset scope:** Only four conditions; no cross-domain or cross-hospital validation.
2. **Environment artifacts:** Some requested tests are unavailable, possibly biasing learning.
3. **Weak baselines:** Comparison methods are not fully aligned in modality or setting.
4. **Preprocessing bias:** Summarization of clinical notes may distort reasoning cues.
5. **Safety unaddressed:** No human-in-the-loop validation or fail-safe mechanism.
6. **Ablation granularity:** Lacks exploration of cyclic vs. joint training and reward sensitivity.

**Questions:**

1. How robust is LA-CDM when scaling to new diseases or hospital systems?
2. What is the impact of unavailable tests on exploration and policy stability?
3. Why does cyclic training outperform joint optimization—any gradient conflict analysis?
4. Does better hypothesis calibration improve decision timing and stop behavior?
5. How sensitive is the cost trade-off to regional price variations?
6. Can hard safety constraints be added to the RL objective?
7. Have clinicians evaluated qualitative reasoning traces for plausibility?

---

> ### Author Response · Authors · 2025-11-21
> **Rebuttal by Authors (Part 1)**
>
> We thank the reviewer for their feedback. We appreciate their comments on our “innovative design,” in which the “two-agent division closely reflects clinical reasoning loops.” We also acknowledge their recognition of our “cost-sensitive optimization” that “demonstrates efficiency improvements without sacrificing accuracy,” and the way our approach “integrates notes, labs, and imaging text within a unified environment.” Finally, we appreciate their note that our “clear ablations” empirically validate each module’s contribution. In the following, we will address all your open questions:
>
> **1. On the dataset**
>
> The MIMIC-CDM dataset contains a subgroup of patients reporting to the emergency department with abdominal pain. It thus represents a challenging and realistic scenario, where doctors must perform differential diagnosis between a narrowed down set of diseases with overlapping symptoms. This step-wise reduction of possible diagnoses is common practice in medicine, making the dataset a meaningful proof of concept for demonstrating clinical decision-making and differential diagnosis. The challenging aspect of differential diagnosis is not differentiating between a broken bone and appendicitis but differentiating between related conditions with similar symptom presentation. Previous studies [1] of LLMs evaluated in zero-shot out-of-distribution settings have shown unsatisfactory results. We propose for the first time a formulation enabling task-specific fine-tuning of LLMs for clinical decision-making. There are no other comparable public datasets available from other hospitals to evaluate cross-hospital generalization.
>
> [1] Hager et al. "Evaluation and mitigation of the limitations of large language models in clinical decision-making." Nature medicine 30.9 (2024): 2613-2622.
>
> **2. Missing requested tests**
>
> As our data does not contain test results for each test and each patient, our method is constrained to the set of clinically valid tests and observed diagnostic pathways contained in the dataset. As noted in the limitations section, the model can therefore only explore a limited spread of testing pathways and is restricted to learning efficiency improvements within testing protocols that were actually performed by doctors, e.g., by replacing or skipping unnecessary tests. In this sense, our objective is not to discover novel clinical strategies, but to optimize decision-making within realistic, expert-demonstrated behavior. We have expanded our limitations section to reflect this.
>
> **3. Choice of baselines**
>
> Our work is the first to attempt fine-tuning clinical decision making models on varied data sources, like clinical notes, lab values and imaging reports. As there are very few prior works on interactive and iterative clinical decision making (especially trained approaches) we had to evaluate SM-DDPO, a work designed to work only with tabular data, in the more comprehensive and realistic setting we are investigating. We did this by providing SM-DDPO with all data that it can process and leaving out non-tabular data. Under these constraints, our comparison is fair and transparently described in the paper. The lack of strong baselines therefore stems from limited prior work.
>
> **4. Preprocessing of clinical notes**
>
> The summarization of clinical notes is necessary to fulfill our academic computational requirements, as overly long texts increase GPU memory use and processing speed. However, we want to stress that the summarization happens without any auxiliary patient information, most importantly without providing the patient disease labels. We have clarified this in our revised manuscript in appendix C. Any lost information, which is possible to occur in any summarisation, therefore potentially only leads to underestimating the performance of our method compared to if full clinical notes could be used.
>
> **5. Potential for humans-in-the-loop**
>
> We thank the reviewer for raising the important point of safety. LA-CDM is explicitly designed as a clinical decision support system and is not intended to act autonomously. In any real-world scenario, a human clinician would remain fully in control of whether a suggested test is ordered or not. If a clinician considers a requested test inappropriate, unsafe, or unnecessary, it can simply be marked as unavailable to the system, without any change to the underlying methodology. Likewise, any additional information obtained independently by the clinician can be incorporated by adding it to the current patient state, allowing the system to continue operating in a human-supervised loop. We will clarify this human-controlled decision process and highlight potentials for humans-in-the-loop in the revised manuscript.
>
> **6. Ablation of cyclic vs. joint training**
>
> Due to limitations of computational resources, joint training of the three objectives is not feasible. We therefore could not explore a joint training approach.

---

> > ### Author Response · Authors · 2025-11-21
> > **Rebuttal by Authors (Part 2)**
> >
> > **7. Impact of hypothesis training**
> >
> > As can be seen in Table 3, the inclusion of the hypothesis agent improves diagnostic performance of the model with F1 micro and macro scores increasing from 81.7 to 84.3 and 78.6 to 82.4, respectively. We do not observe a strong difference in cost efficiency between both settings.
> >
> > **8. Sensitivity to cost variations**
> >
> > We thank the reviewer for suggesting this interesting evaluation. In order to evaluate cost table sensitivity, we performed two additional experiments.
> >
> > 1) *Inter-institutional cost variation*
> >
> > To evaluate sensitivity to institutional cost variations, we simulate a potential other hospital’s cost distribution by performing slight variations to our cost table. After these variations, the model’s performance and cost efficiency remains largely unaffected. Details of this experiment can be seen in the appendix of the revised manuscript.
> >
> > 2) *Sensitivity to specific test cost differences*
> >
> > In order to test model reactance to specific cost signals, we construct an artificial test table where one test’s cost, the complete blood count, has been increased to be very expensive. After retraining with this adapted cost table, we observe that the test is requested far less often. Under normal prices, the test is requested in 19.1% of cases, when it is more expensive this rate reduces to 2.5% of cases.
> >
> > Details on both experiments are provided in the appendix D of the revised manuscript.
> >
> > Thank you again for your feedback. We hope our reply clarifies your open questions and will be helpful for your final evaluation.

---

> > > ### Author Response · Authors · 2025-11-28
> > >
> > > Thank you again for constructive feedback which helped further improve the paper. Given the limited remaining time of the discussion phase, we would kindly ask whether you have any remaining concerns.

---

### Official Review · Reviewer_sB6P · 2025-11-01

**Soundness:** 3
**Presentation:** 3
**Contribution:** 3
**Rating:** 4
**Confidence:** 3

**Summary:**

This paper proposes LA-CDM, a two-agent system combining hypothesis generation and decision-making modules trained with hybrid supervised and reinforcement learning for iterative clinical diagnosis.

**Strengths:**

1. The paper addresses an important gap in clinical AI systems by modeling the iterative nature of diagnostic reasoning.
2. The paper provides thorough comparison against multiple baselines and includes cost-efficiency metrics.
3. The paper attempts to integrate uncertainty estimation with sequential decision-making in a clinical context.
4. The paper includes detailed prompts and implementation details that facilitate understanding of the approach.

**Weaknesses:**

1. The paper introduces separate hypothesis and decision agents without sufficiently justifying their architectural separation. Specifically:
- The hypothesis agent generates diagnostic hypotheses and confidence scores, while the decision agent uses this output to select tests or final diagnoses. However, both agents inherently engage in diagnostic reasoning, and the decision agent could potentially internalize hypothesis generation and confidence estimation.
- The confidence score produced by the hypothesis agent could be implicitly learned by the decision agent through its reasoning process, eliminating the need for explicit separation. This raises questions about whether the dual-agent design introduces unnecessary complexity.

2. The reinforcement learning component is trained on retrospective clinical data, which poses significant off-policy learning challenges:
- Historical data reflects actions taken by clinicians, which may not align with the optimal policy learned by the agent. This can lead to distributional shift and biased policy evaluation, as the agent may propose test sequences not represented in the data.
- The paper mentions handling unavailable tests by prompting the agent to choose alternatives, but this does not address the core issue of learning from a fixed dataset with limited action coverage. Off-policy correction methods (e.g., importance sampling, conservative Q-learning) or imitation learning techniques are not discussed.

**Questions:**

See above

---

> ### Author Response · Authors · 2025-11-21
> **Rebuttal by Authors**
>
> We thank the reviewer for their feedback. We appreciate their assessment that our work “addresses an important gap in clinical AI systems by modeling the iterative nature of diagnostic reasoning,” that it “provides thorough comparison against multiple baselines and includes cost-efficiency metrics,” and that we include “detailed prompts and implementation details that facilitate understanding of the approach”. In the following, we will address all your open questions:
>
> **1. Separation of hypothesis and decision agent**
>
> We would like to clarify that the two agents share the same LLM and model weights during the whole training procedure and only differ in their prompts and objectives. Consequently, the agents can already implicitly benefit from all tasks during training. Importantly, the separation is not a source of architectural complexity, but rather a step to reduce complexity during training. Collapsing all three objectives (hypothesis generation, uncertainty estimation, and decision policy learning) into a single agent would require a more complex training procedure, involving, e.g., loss masking and step-wise generations, than separating them into different LLM generations.
>
> Moreover, relying on the decision agent to implicitly internalize hypothesis generation and uncertainty estimation is empirically worse than the explicit modeling. Our ablation results in Table 3 demonstrate that the explicit hypothesis agent leads to improved diagnostic performance.
>
> **2. Learning from retrospective data and off-policy learning**
>
> We thank the reviewer for highlighting this important consideration and would like to clarify our reinforcement learning setting. While our environment is constructed from retrospective clinical data, our reinforcement learning algorithm is not off-policy in the classical sense of offline RL. In our GRPO training loop, we do not optimize directly on clinician trajectories. Instead, at each update step we roll out the current policy on a log-based simulator: for a given patient, the policy chooses a sequence of tests and a final diagnosis; the environment then returns either the recorded test result (if available in the chart) or a special "not available" observation, plus a reward based on test costs and diagnostic correctness. GRPO is applied to these trajectories that were generated by the current policy itself, i.e., the behavior and target policies coincide at every update. As a result, our policy-gradient estimates are on-policy, and classical off-policy corrections such as importance sampling or conservative Q-learning are not required for the correctness of the GRPO update. We clarified our reinforcement learning setting in the updated manuscript.
>
> Our method is indeed constrained to the set of clinically valid tests and observed diagnostic pathways contained in the dataset. As noted in the limitations section, the model can therefore only explore a limited spread of testing pathways and is restricted to learning efficiency improvements within testing protocols that were actually performed by doctors, e.g., by replacing or skipping unnecessary tests. In this sense, our objective is not to discover novel clinical strategies, but to optimize decision-making within realistic, expert-demonstrated behavior. We have expanded our limitations section to reflect this.
>
> Thank you again for your feedback. We hope our reply clarifies your open questions and will be helpful for your final evaluation.

---

> > ### Author Response · Authors · 2025-11-28
> >
> > Thank you again for constructive feedback which helped further improve the paper. Given the limited remaining time of the discussion phase, we would kindly ask whether you have any remaining concerns.

---

### Author Response · Authors · 2025-12-03
**Summary of the rebuttal process**

Dear Area Chair,

We thank the reviewers for their time and valuable feedback. Given the special situation, and, we want to take this opportunity to provide a short summary of the reviews, the discussion with reviewer epNq, and the resulting changes and additions to our work.

The key concerns discussed in the reviews were:

1. Test cost sensitivity (j7xL, epNq)
2. Dataset generalizability (j7xL, epNq)
3. Learning from retrospective data (sB6P, j7xL, epNq)
4. Preprocessing of clinical notes (j7xL, epNq)

In our rebuttal and revised manuscript, we responded as follows:

1. We performed **two further experiments** to evaluate the test cost sensitivity of our method. Under **small variation of costs**, simulating the cost table of different hospitals, we found no significant difference between model performance and test efficiency, indicating robustness to slight variations. In a second experiment, we **artificially increase the cost of one particular test**, Complete Blood Count (CBC). In this setting, the prevalence of CBC requests are starkly reduced. **Reviewer epNq acknowledged this concern as resolved**.

2. We clarified that the dataset is the **only publicly available dataset** for evaluating the **realistic setting of iterative differential diagnosis**. As doctors usually have to differentiate between similar diseases with similar disease expressions, this dataset presents a **challenging proof-of-concept**. We acknowledge that the scope of the dataset is limited. However, we agree with reviewer epNq’s comment that this is a common problem in this area, but believe this research direction to be valuable, nonetheless.

3. We clarified that our method **can only explore a clinically valid subset of tests per patient** that were previously requested by a doctor. The aim of our method is therefore not to find novel testing strategies but to **become more efficient within established testing pathways**.

4. We clarified that the processing of clinical notes is done **due to compute limitations**, as the notes without summarization would be too long for our academic computational requirements. The **summarization is done without any auxiliary information** or ground truth labels. **Reviewer epNq acknowledged this concern as resolved**.

We believe these additional analyses and clarifications substantially strengthen the paper and directly address the main concerns raised in the initial reviews. We again thank the area chair and the reviewers for their efforts.

Best regards

The authors

---

### Meta-Review · Area_Chair_Uv1w · 2025-12-29

**Summary:**

The paper received mixed scores. However, the rejects are weak, stating that they would not mind if the paper is accepted, and overall the reviews are positive. The primary criticism is about the studied dataset and not about the method. However, obtaining high quality medical datasets is challenging, yet a positive and important direction that is welcomed at ICLR. The authors successfully addressed technical concerns (cost sensitivity, RL robustness) during the rebuttal. I recommend acceptance at this point.

**Reviewer Concerns:**

no concerns

**Reviewer Scores:**

not relevant

---

### Decision · Program_Chairs · 2026-01-26

Accept (Poster)